# Finite Element Analysis of Self-Healing Concrete Beams Using Bacteria

**DOI:** 10.3390/ma15217506

**Published:** 2022-10-26

**Authors:** Ghada G. Salem, Vera V. Galishnikova, S. M. Elroba, Nikolai I. Vatin, Makhmud Kharun

**Affiliations:** 1Department of Construction Engineering, Egyptian Russian University (ERU), Badr City 11829, Egypt; 2Department of Civil Engineering, Moscow State University of Civil Engineering (National Research University), 26 Yaroslavskoye av., 129337 Moscow, Russia; 3Department of Civil Engineering, Peoples’ Friendship University of Russia, 6 Miklukho-Maklaya st., 117198 Moscow, Russia; 4Department of Civil Engineering, Peter the Great St. Petersburg Polytechnic University, 195251 St. Petersburg, Russia

**Keywords:** reinforced concrete beams, ANSYS models, bacterial concrete, self-healing concrete, *Bacillus subtilis*

## Abstract

Deterioration or crack formation in concrete elements is a phenomenon that cannot be easily avoided, and it has a high cost of repair. A modern technology that needs wider study is the use of the bio-precipitation of calcium carbonate using bacteria to increase a structures’ capacity. The current research presents an analytical study on self-healing concrete beams using bacteria to enhance the beam’s capacity. A Finite Element Analysis on (ANSYS 15.0) was carried out to study the effect of the bacteria concentration (the weight of bacteria to cement weight 1%, 2%, and 3%), the type of bacteria (*Bacillus subtilis, E. coli,* and *Pseudomonas sps.*), and the loading (a one-point load, a two-point load, and a distributed load on four points) on concrete beams. Two beams were chosen from previous experimental research and simulated on the ANSYS before carrying out our parametric study to verify the validity of our simulation. Following this, our parametric study was carried out on eight beams; each beam was loaded gradually up to failure. The results show that the optimum type of bacteria was the *Bacillus subtilis*, and that the bacteria concentration of 3% for *Bacillus subtilis* can increase the beam’s capacity by 20.2%. Also, we found that distributing the load to four points led to the increase of the beam’s capacity by 74.5% more than the beam with a one-point load.

## 1. Introduction

The use of self-healing bacterial concrete in real applications became a topic of interest due to its ability to extend the service life of structures. Considerable research has been undertaken on the mechanical properties of bacterial concrete cubes and cylinders, but few studies have applied this technique to the reinforced concrete elements on a large scale such as beams, columns, and slabs, etc. Therefore, we still need to study the matter more extensively with different parameters.

The use of self-healing concrete can prove its efficiency as smart material, designed and inspired by nature such as in skin tissue or bone structure [1,2]. Nathalie K. Guimard et al. presented the most recent and promising advancements in the field of self-healing materials [3]. Stijn Billiet et al. showed the difference between extrinsic materials and intrinsic materials, where the extrinsic materials obtain the self-healing property by adding healing agents, while the self-healing property is achieved by the material itself in intrinsic materials [4].

Microcracks can create a direct pathway for undesired substances to penetrate concrete elements, causing reinforcement corrosion and requiring costly manual maintenance and repair. These cracks affect the structure’s functionality, and they also affect the durability and strength of the structure [5,6]. An increase in microcracks leads to increased permeability which makes the structure more sensitive to the applied loads [7,8]. Biqin Dong et al. presented the development of self-healing materials that show promise for the permeability healing of concrete or other cementitious composites [9]. Ghasan Fahim Huseien et al. studied the self-healing of microcracks with polymeric admixtures and showed that the optimum polymer to cement ratio was determined to be 10% [10]. Autogenous concrete can also heal these microcracks, but this technique is ineffective for a crack width of more than 0.06 mm [11]. Therefore, to enhance concrete resistance to these defects and degradations, the innovation of self-healing concrete using bacteria is promising, and has the following advantages: enhancing the concrete element’s capacity, healing cracks for a long period (more than 15 years), being more economical, enabling access to all locations, and being eco-friendly [12]. The formation of CaCO_3_ by the induced bacteria to heal microcracks has also been investigated through many other studies [13,14]. H. M. Jonkers et al. carried out the compressive strength test on cubes of cement stone with the dimensions of 4 × 4 × 4 cm, and the splitting-tensile strength test on cement stone cylinders with the dimensions of 2.2 cm diameter and 3 cm height, to study the effect of adding bacteria and organic substances on cement paste strength [15]. H. M. Jonkers in another research study investigated the self-healing capacity of pre-cracked concrete slabs sawed from concrete cylinders that had been water-cured for 56 days (2 months); the slabs were 10 cm diameter and 1.5 cm thickness [16]. Mian Luo et al. carried out a permeability test on cylindrical specimens that were 75 mm diameter and 35 mm height, and they also created prismatic specimens with dimensions of 40 × 40 × 160 mm that were used for the image characterization of crack healing efficiency [17]. Wasim Khaliq et al. presented the process of the crack healing phenomenon in concrete on cylindrical specimens by microbial activity of the *Bacillus subtilis* bacteria [18]. C. Lors et al. showed that the healing of autogenous and carbonated microcracks by precipitating calcium carbonate can be efficient due to an adapted bacterial suspension [19]. N. H. Balam et al. studied the use of microbial carbonate precipitation in Light Weight Aggregate Concrete and carried out the compressive strength test and permeability test on standard cubes and cylinders to determine their mechanical properties [20]. Varenyam Achal et al. studied the role of the *Bacillus* sp. bacteria on the durability properties and remediation of cracks in cementitious structures [21]. A. Faisal Alshalif et al. studied the isolation of sulphate reduction bacteria (SRB) to improve the compressive strength and water penetration of bio-concrete on standard cubes with a size of 150 × 150 × 150 mm; and the results showed that use of (SRB) in concrete can increase compressive strength and decrease water penetration [22]. M. O’Connell et al. studied the effect of sulfate/sulfuric acid on concrete with particular reference to the role of sulfate-reducing and sulfur-oxidising bacteria [23]. Yusuf et al. in their tests used NO_3_-reducing bacteria to enhance crack closure performance of microbial mortar [24] and to develop corrosion resistant self-healing concrete [25]. Wiktor et al. showed that Bacillus alkalinitrilicus and nitrate-reducing bacteria could heal up to a 0.46 mm crack width [26]. Mohammad et al. studied the effect of sustained service loads on the self-healing bacterial concretes using Sporosarcina pasteurii bacteria on beams 120 × 150 × 1200 mm, and they showed that adding bacteria and nutrients to the concrete decreases the water absorption, therefore, increasing the flexural capacity of beams [27]. Shashank B.S et al. studied the behaviors of cracks under sustained loading conditions on 15 reinforced concrete beams with the dimensions of 150 × 200 × 1500 mm and they concluded that use of bacteria improves the fracture behavior of concrete [28].

Only a small number of numerical studies have been carried out to explain the self-healing process of cementitious elements, such as I. Rohini et al. who added the *Bacillus subtilis* to reinforced concrete beams to examine the flexural behavior of bacterial demolition waste concrete; their experimental results were validated with the use of the Finite Element Software Abaqus [29]. Aliko-Benítez et al. introduced autogenous self-healing concrete modeling in their numerical research [30]. Also, S.V. Zemskov et al. introduced a mathematical self-healing model to describe the bacteria-based self-healing process which depended on organic acids oxidation [31].

With reference to previous research it can be concluded that adding bacteria to the concrete mix for cubes, cylinders, and prismatic specimens has a significant effect on enhancing their mechanical properties;however, we still need to apply this technology to the reinforced concrete elements on a large scale such as to beams, columns, and slabs, etc. Therefore, the current research is an investigation into adding bacteria to reinforced concrete beams by changing three different parameters on software package (ANSYS 15.0):Bacteria concentration (1%, 2% and 3%);Type of bacteria (*Bacillus subtilis, E. coli* and *Pseudomonas sps.*);Case of loading (One-point load, two-point load, and distributed load on four points).

The Finite Element Models on ANSYS APDL are verified initially by two experimental beams previously tested by Milan Joy et al. [32], followed by our parametric study.

## 2. Materials and Methods

### 2.1. Modeling Assumptions

The reinforced concrete beams were simulated on ANSYS APDL using three dimensional elements: SOLID65, LINK180, and SOLID185 for concrete, reinforcement, and steel plates, respectively. The mechanical properties for the concrete elements were assumed to be changed during the solution for each model (except the control specimen which did not contain bacteria) to simulate the activation of bacteria, which happened in the laboratory for the actual beams when the bacteria received nutrition due to the opened cracks. The mechanical properties of bacterial concrete could be graduated through the modelling from those of concrete without bacteria to those of concrete with bacteria using the command (MPCHG), after reaching the load step in which the cracks begin to appear.

### 2.2. Description of Elements

SOLID65 was used to model concrete. The element was defined by eight nodes having three degrees of freedom at each node: translations were in the nodal x, y, and z directions. The solid was capable of cracking in tension and crushing in compression. A 3-D element (LINK180) was used to model the reinforcement. The element was a uniaxial tension–compression element with three degrees of freedom at each node: translations were in the nodal x, y, and z directions. Tension-only (cable) and compression-only (gap) options were supported. A 3-D solid element (SOLID185) was used to model the steel plates for loading and support. Figure 1 illustrates the types of elements.

### 2.3. Verification Models

The aim of this section is to verify the validity of our simulation on ANSYS, and not to repeat the previous work. Therefore, after the verification, we could build an analytical study based on our own parameters. Two reinforced concrete beams from previous experimental work by Milan Joy et al. [32] were chosen to verify our ANSYS results. The research aimed to find the effect of bacterial concrete with fly ash on the strength characteristics of concrete. The beam’s dimensions were 230 × 300 × 1500 mm and it was reinforced by three bars of a 16 mm diameter as the main reinforcement at the bottom and two bars that were 12 mm in diameter at the top. The stirrups were two-legged and 10 mm diameter @ 120 mm. The concrete dimensions and reinforcement details are shown in Figure 2.

Two beams from Milan’s experimental work [32] were chosen to be the verification models:Control Specimen (CS);Bacterial concrete beam with fly ash (BCFA).

The bacteria species *Bacillus subtilis* was selected in Milan’s research. The mechanical properties of concrete with and without bacteria and fly ash were obtained. The compressive strength was obtained from cubes of 150 mm on each side and the split tensile strength was obtained from cylinders of 150 mm in diameter and 300 mm in height. Table 1 shows these mechanical properties.

Figure 3 illustrates the test set-up for beams in the laboratory with two-point loads and Figure 4 illustrates our simulation of beams on ANSYS with the same loads and boundary conditions. Each beam was loaded gradually up to failure then the load-displacement curves were obtained and compared with the experimental curves.

### 2.4. Parametric Study

A parametric study was carried out on eight beams with the same concrete dimensions and reinforcement details as the previous verification models. The eight beams were divided into three groups.

Group (1) addressed the first parameter (bacteria concentration), with all other parameters kept constant except the bacteria concentration for the purposes of comparison. The bacteria concentration was the weight of the bacteria to the weight of cement, and it differed between 1%, 2%, and 3% for the same type of bacteria (*Bacillus subtilis*)—the most popular of all types. It was the same case for loading (two-point load). A comparison was then made with the conventional beam (the control beam without bacteria).

Group (2) addressed the second parameter (the type of bacteria). In this group, we began to compare the effect of adding *Bacillus subtilis* with the effect of adding other bacteria types (*E. coli* and *Pseudomonas sps.*) with the bacteria concentration remaining constant at 3% and the loading at a two-point load. Comparisons were made with the control specimen.

Group (3) addressed the third parameter (case of loading) and its effect on the capacity of beams with a constant bacteria concentration; the type of bacteria was 3% of *Bacillus subtilis*. Three cases of loading were used: a one-point load, a two-point load, and distributed loads on four points.

All beams were simulated on ANSYS using the same elements: SOLID65, LINK180, and SOLID185 for concrete, reinforcement, and steel plates. Table 2 summarizes the specimens’ labels and details.

Table 3 illustrates the mechanical properties of the concrete that was used in our models and these were obtained from the compressive strength tests and tensile strength tests from K. Vijaya et al. [33].

## 3. Results and Discussion

The failure loads were obtained, and the displacements were measured at the mid-span of beams for each load step, then the load-displacement curves were drawn. The crack patterns and modes of failure were also illustrated.

### 3.1. Results of the Verification Models

A comparison between ANSYS results and the experimental results was carried out to validate our finite element models. The failure loads and the corresponding displacements showed a good agreement, as illustrated by the percentage of ANSYS failure load (P_ans_) to experimental failure load (P_exp_) as follows:(P_ans_)/(P_exp_)% for the control specimen (CS) = 91.4%;(P_ans_)/(P_exp_)% for the bacterial concrete beam with fly ash (BCFA) = 94.5%.

The numerical curve showed the nonlinearity of the concrete material as it is defined on ANSYS, therefore it took the shape of a parabola, showing a small gap in the elastic zone between both curves. Figure 5 shows the load-displacement curves and Table 4 illustrates the failure loads and the corresponding displacements.

Figure 6 illustrates the crack patterns of the beams. Cracks in the green and blue colors are critical and will cause failure. It is evident from the crack patterns among the load steps for each beam up to failure, that the mode of failure in the control specimen was splitting shear failure, while the mode of failure in the BCFA was diagonal tension failure. The vertical cracks (flexural cracks) in the BCFA, shown in green and blue, began at the bottom of the beam, then grew both in width and length and bent in a diagonal direction as they moved to the upper part of the beam toward the loading point.

### 3.2. Results of the Parametric Study

The parametric study results will be discussed in the current section according to the effect of each parameter.

#### 3.2.1. Concentration of Bacteria

Three concentrations of bacteria were studied (1%, 2%, and 3%) for the same type of bacteria (*Bacillus subtilis*). Figure 7a shows that the increase in bacteria concentration could enhance the beam’s capacity, depending on its percentage. The results show that percentages of 1% and 3% concentration could enhance the beam’s capacity by 16.2% and 20.2%, respectively, compared to the control specimen. A concentration of 2% resulted in no difference when it was compared to a concentration of 1%, as it increased the capacity by 17%, which was approximately the same amount.

From Figure 7b, it is evident that an increase in bacteria concentration led to a decrease in the deflection of beams, which could be observed by reading the deflection value at the same load step for all beams. Figure 7b shows the deflection value for all beams at the same load of 286.5 KN, which is the failure load for the control specimen. The results show a decrease in deflection at the mid-span of the beams by 2%, 14.8%, and 15.7% for bacteria concentrations of 1%, 2%, and 3%, respectively.

#### 3.2.2. Type of Bacteria

As shown in the previous parameter, a percentage of 3% is more effective for enhancing the beam’s behavior. Therefore, in the current section, three types of bacteria were investigated (*Bacillus subtilis, E. coli*, and *Pseudomonas sps.*) at the same concentration of 3%. The following bar chart in Figure 8a illustrates the failure loads.

The results show that *Bacillus subtilis* was more effective than the other types with a constant percentage, as seen in the increase in the capacity of the beams by 20.2%, while the *E. coli* and *Pseudomonas sps.* increased the capacity of the beams by 16.1% and 14.9%, respectively. Also, the deflection of the beams at the same load could be decreased by using the *Bacillus subtilis* bacteria over the other types as shown in Figure 8b. The *Bacillus subtilis* decreased the deflection by 15.7%, while the *E. coli* decreased the deflection by 4.7% only, at the same bacteria concentration of 3%. Conversely, the *Pseudomonas sps.* with a concentration of 3% was unable to decrease the deflection at all.

#### 3.2.3. Loading Cases

Three loading cases were installed for the beams in Group (3) with a constant bacteria concentration and type of bacteria. The results show that distributing the load to many points enhanced the beam’s capacity. The beam with the distributed load on four points had a higher capacity by 74.5% than the beam with a one-point load and a higher capacity by 13% than the beam with a two-point load. The following bar chart in Figure 9a illustrates the failure loads.

Figure 9b shows that distributing loads on more points can decrease the deflection of the beams, as the deflection of the beam with a distributed load on four points decreases by 77.8% when it is compared to the deflection of the beam with a one-point load. Figure 10 illustrates the load-Displacement curves for beams in Group (3).

The crack patterns are illustrated in Figure 11 and the gradation of load steps show that the mode of failure is a diagonal tension failure. The green and blue cracks began as vertical cracks (flexural cracks) due to flexural tensile stress and grew both in width and length with the increase of load; then they bent in a diagonal direction as they moved to the upper part of the beam toward the loading point.

The bottom reinforcement in all beams in each parameter reached the yield stress before the concrete failed due to a ductile failure, which is desireable in the beam design process. Figure 12 shows the stresses in reinforcement for one beam as an example.

## 4. Conclusions

According to the current analytical study on the ANSYS software package, the following conclusions are obtained:The ANSYS failure loads (P_ans_) represented 91.4% to 94.5% of the experimental failure loads (P_exp_) which is good agreement with the verification models:
a.(P_ans_)/(P_exp_)% for the control specimen (CS) = 91.4%b.(P_ans_)/(P_exp_)% for the bacterial concrete beam with fly ash (BCFA) = 94.5%;
Adding bacteria to the reinforced concrete beams increased their capacity and decreased the deflection, depending on the bacteria type and concentration;The *Bacillus subtilis* was the optimum type of bacteria and had a higher significant effect than the E. coli and *Pseudomonas sps.* especially with a concentration of 3%, as it increased the capacity of the beam by 20.2% and decreased the deflection by 15.7%, compared to the control specimen without bacteria;Adding 1% and 2% of *Bacillus subtilis* to the reinforced concrete beams had approximately the same effect on the beam’s capacity, which increased by 16.2% and 17%, respectively, compared to the control specimen without bacteria;Adding 1% *Bacillus subtilis* to the reinforced concrete beam had no effect on deflection, as it decreased the deflection by 2% only;The use of *Pseudomonas sps.*, even with a concentration of 3%, was unable to decrease the deflection at all;Distributing the load on many points enhanced the beam’s capacity to support more than a concentrated load on one or two points, as the beam with the distributed load on four points had a higher capacity by 74.5% than the one-point loaded beam, and by 13% than the two-point loaded beam. Also, the deflection decreased for the beam with a distributed load on four points by 77.8% when it was compared to the one-point loaded beam, and 67.6% when it was compared to the two-point loaded beam.

### Future Directions and Limitations

A greater number of parameters could be studied to understand more widely the self-healing nature of concrete using bacteria and to discover their benefits. Also, the study could be repeated for other types of elements such as columns, slabs, and frames, etc., instead of beams, to understand the behavior of these elements under the use of self-healing concrete. The following parameters can be taken into consideration:Low and high temperature;The structures’ scale;Carrying out an experimental study;Changing the nutrition type for bacteria;Using capsules to preserve bacteria inside the concrete until cracks occur;Adding bacteria to the marine concrete.

Some limitations may be facing this analytical study such as simulating the nutrition of bacteria or simulating capsules to preserve bacteria inside the concrete until cracks occur. Therefore, these two parameters need further experimental study to assess their potential implementation.

## Figures and Tables

**Figure 1 materials-15-07506-f001:**
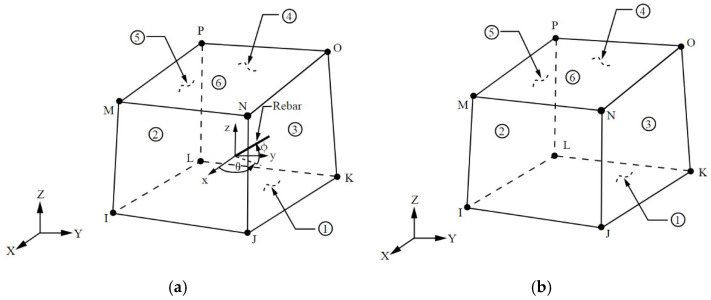
Types of Elements (**a**) SOLID65. (**b**) SOLID185. (**c**) LINK180.

**Figure 2 materials-15-07506-f002:**
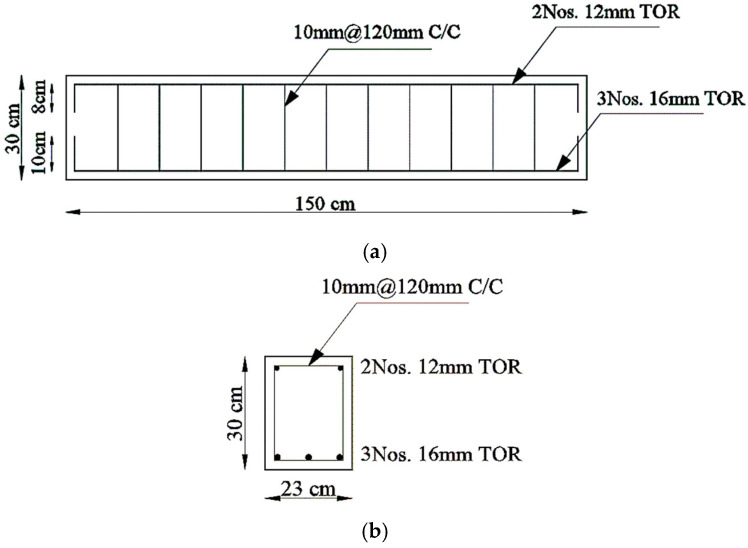
Concrete Dimensions and Reinforcement Details for the Verification Models. (**a**) Elevation. (**b**) Cross-Section.

**Figure 3 materials-15-07506-f003:**
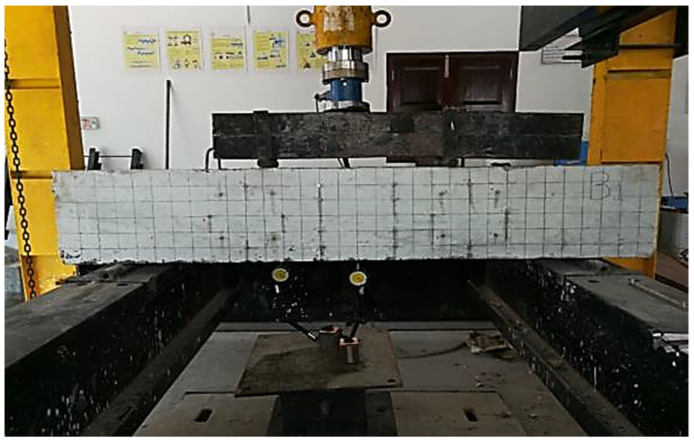
Test Set-Up for the Experimental Beams.

**Figure 4 materials-15-07506-f004:**
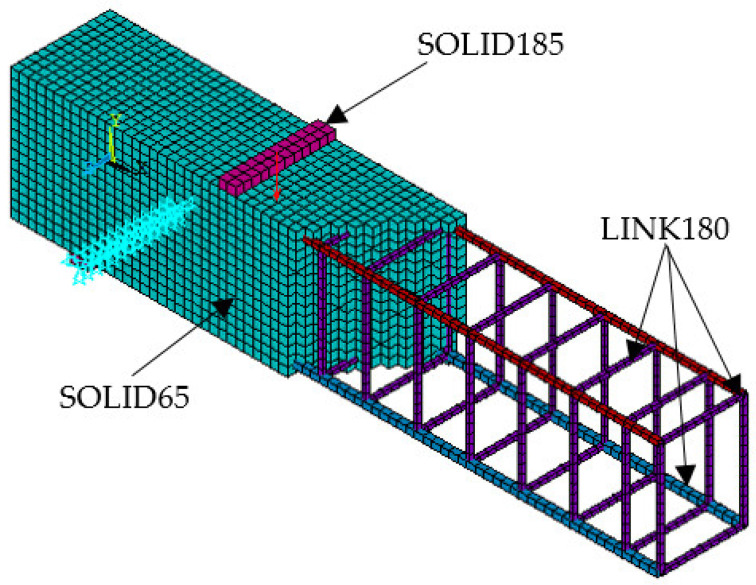
The Simulations of beams on ANSYS.

**Figure 5 materials-15-07506-f005:**
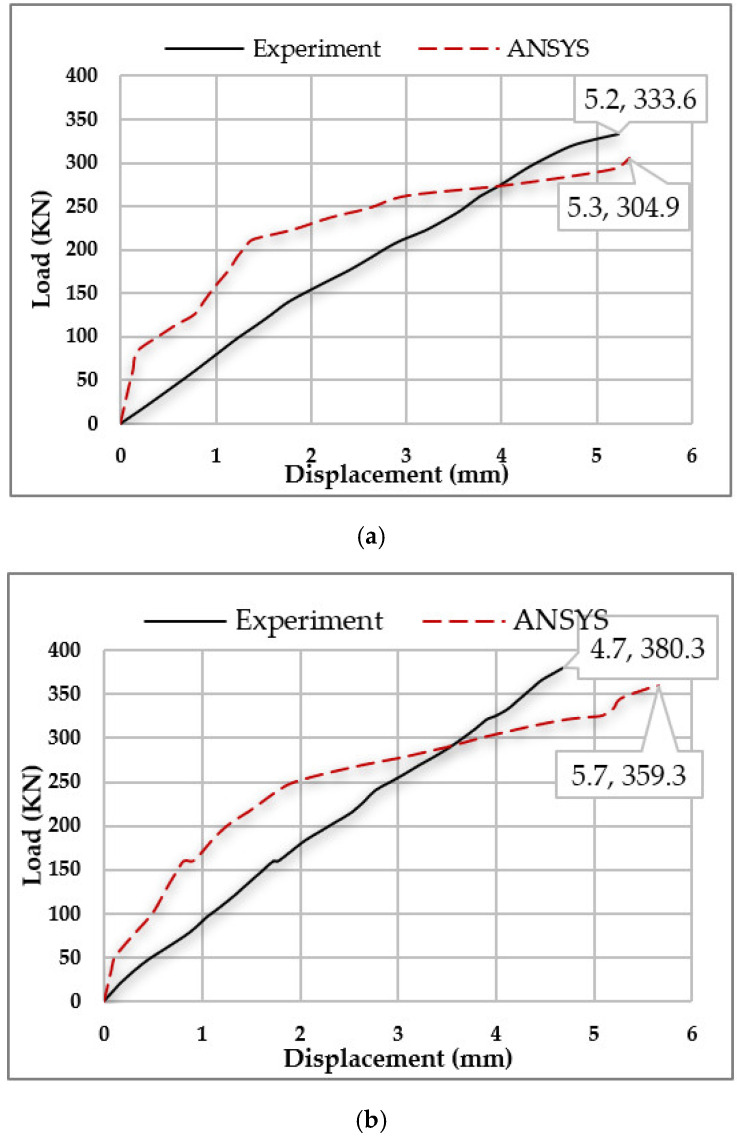
Load-Displacement Curves for Experimental Specimens Vs. Their Simulation on ANSYS. (**a**) Control Specimen (CS). (**b**) Bacterial concrete beam with fly ash (BCFA).

**Figure 6 materials-15-07506-f006:**
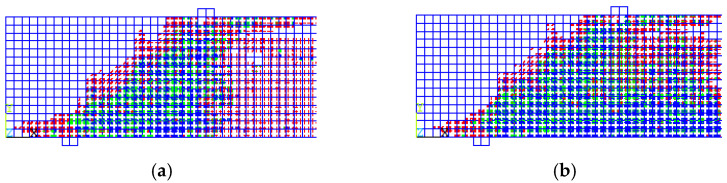
Crack Patterns for the Verification Models. (**a**) Control Specimen at Failure Load = 304.9 KN. (**b**) BCFA at Failure Load = 359.3 KN.

**Figure 7 materials-15-07506-f007:**
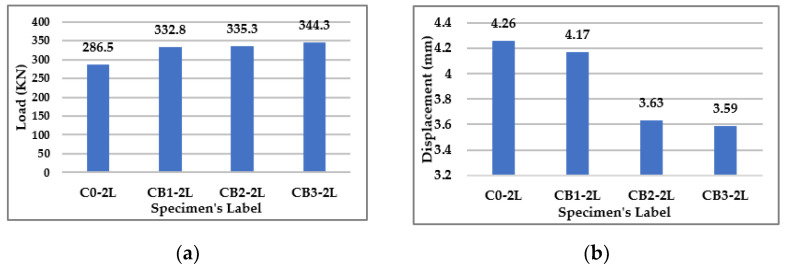
Results for Group (1). (**a**) Failure Loads for Group (1). (**b**) Deflection at the Mid Span of Beams in Group (1) at the same Load = 286.5 KN.

**Figure 8 materials-15-07506-f008:**
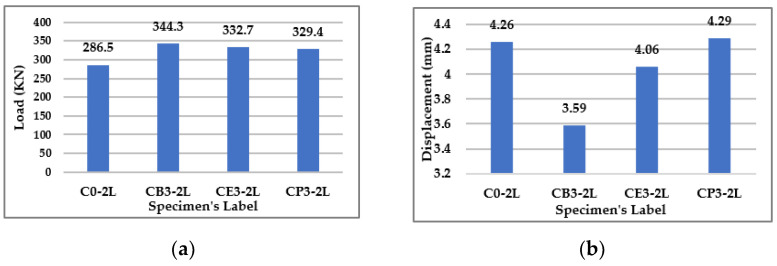
Results for Group (2). (**a**) Failure Loads for Group (2). (**b**) Deflection at the Mid Span of Beams in Group (2) at the same Load = 286.5 KN.

**Figure 9 materials-15-07506-f009:**
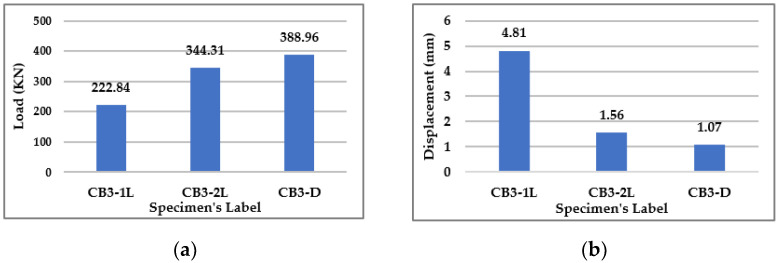
Results for Group (3). (**a**) Failure Loads for Group (3). (**b**) Deflection at the Mid Span of Beams in Group (3) at the same Load = 222.84 KN.

**Figure 10 materials-15-07506-f010:**
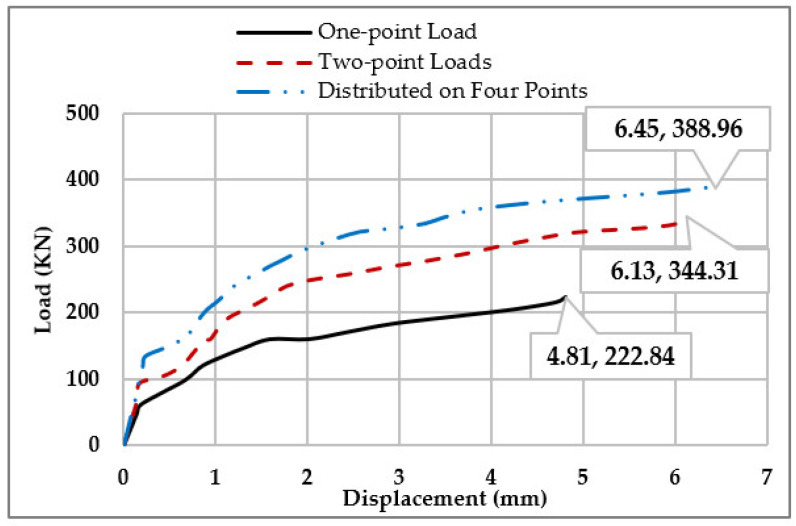
Load-Displacement Curves for Group (3).

**Figure 11 materials-15-07506-f011:**
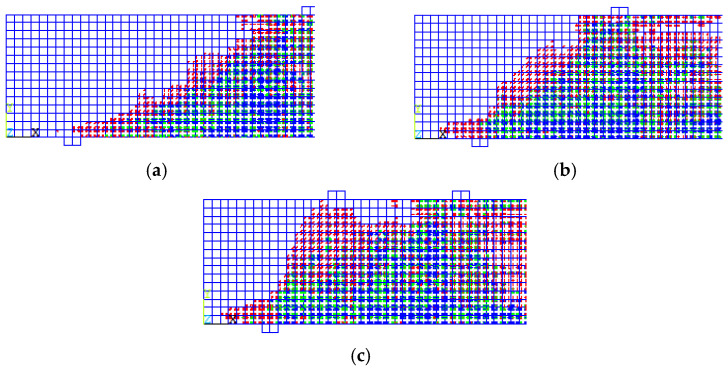
Crack Patterns for Group (3). (**a**) One-Pont Load. (**b**) Two-Point Loads (**c**) Distributed Load on Four Points.

**Figure 12 materials-15-07506-f012:**
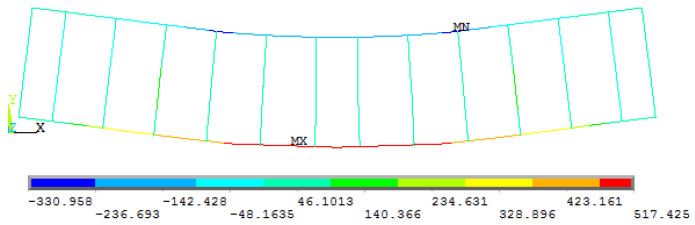
Stresses in the reinforcement for beam CB3-2L.

**Table 1 materials-15-07506-t001:** Mechanical Properties of Concrete for Verification Models.

Specimen Label	Specimen Details	Compressive Strength MPa	Tensile Strength MPa
CS	Control specimen	31.11	3.53
BCFA	Bacterial concrete with fly ash	36.44	3.89

**Table 2 materials-15-07506-t002:** Description of ANSYS Models for Parametric Study.

Group No.	Specimen’s Label	Specimen’s Detail	Type of Bacteria	Bacteria’s Concentration	Case of Loading
1	C0-2L	Control	-	-	Two-Point Loads
	CB1-2L	Bacterial Concrete	*Bacillus subtilis*	1%	Two-Point Loads
	CB2-2L	Bacterial Concrete	*Bacillus subtilis*	2%	Two-Point Loads
	CB3-2L	Bacterial Concrete	*Bacillus subtilis*	3%	Two-Point Loads
2	C0-2L	Control	-	-	Two-Point Loads
	CB3-2L	Bacterial Concrete	*Bacillus subtilis*	3%	Two-Point Loads
	CE3-2L	Bacterial Concrete	*E. coli*	3%	Two-Point Loads
	CP3-2L	Bacterial Concrete	*Pseudomonas sps*.	3%	Two-Point Loads
3	CB3-1L	Bacterial Concrete	*Bacillus subtilis*	3%	One-Point Load
	CB3-2L	Bacterial Concrete	*Bacillus subtilis*	3%	Two-Point Loads
	CB3-D	Bacterial Concrete	*Bacillus subtilis*	3%	Distributed

**Table 3 materials-15-07506-t003:** Mechanical Properties of Bacterial Concrete with different percentages and types.

Type of Bacteria	Bacteria’s Concentration	Compressive Strength MPa	Tensile Strength MPa
Control	-	30	4
*Bacillus subtilis*	1%	33.6	4.09
*Bacillus subtilis*	2%	36.58	4.22
*Bacillus subtilis*	3%	38.95	4.3
*E. coli*	3%	34.67	4.17
*Pseudomonas sps.*	3%	33.74	4.13

**Table 4 materials-15-07506-t004:** Failure Loads and Corresponding Displacements for the Verification Models.

Specimen’s Label	Experimental Failure Load (P_exp_) (KN)	Experimental Displacement (mm)	ANSYS Failure Load (P_ans_) (KN)	ANSYS Displacement (mm)	P_ans_/P_exp_%
CS	333.6	5.2	304.9	5.3	91.4%
BCFA	380.3	4.7	359.3	5.7	94.5%

## Data Availability

Not applicable.

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
