# Peer review of "Finite Element Analysis of Self-Healing Concrete Beams Using Bacteria"

_materials, 2022, doi:10.3390/ma15217506_

Round 1
Reviewer 1 Report (New Reviewer)
In the review of the manuscript titled: Finite Element Analysis of Self-Healing Concrete Beams using Bacteria. The authors have provided a good description and the methodology is also fine. I would like to see this review publish but after some questions as follow;
1. Why did the authors just focus on the Finite Element Analysis of the software package?
2. The authors are requested to compare the results with previously reported results in order to verify our ANSYS results.
3. Why the bacteria of the species Bacillus Subtilis were selected in Milan's research?
4. Why did the authors fix the bacteria’s concentration at 3% with the same case of loading (two-point loads)?
5. The authors stated that the numerical curve shows the nonlinearity of the concrete material as it is defined on ANSYS. Why the numerical curve is nonlinear?
Author Response
Why did the authors just focus on the Finite Element Analysis on software package?
- Authors’ reply: Because the current research is an analytical study only, for the self-healing concrete beams using the Finite Element Method, but not an experimental study which can be a search point for another research. Also, in the current importance nowadays of using artificial intelligence, we want to get the benefits that structural analysis programs offer to us in our research.
The authors are requested to compare the results with previously reported results in order to verify our ANSYS results.
- Authors’ reply: The comparison is already done before carrying out the parametric study to assure that the author is expert in using the ANSYS software package, in order to give credibility of the parametric study’s results. Therefore, according to the beams we will be exposed to later in the parametric study, we need to prove this on two types of beams:
- Reinforced concrete beam without bacteria to be the control specimen.
- Reinforced concrete beam with bacteria.
Which was happened in the section of verification models.
Why the bacteria of the species Bacillus Subtilis were selected in Milan's research?
- Authors’ reply: Research on microbial concrete has predominantly considered spore-forming bacteria, such as from the Bacillus species, as they can survive under unfavorable conditions of high alkalinity and low water availability by forming spores. Consequently, when such conditions are prevalent, the microbial induced calcite precipitation of spore-forming bacteria is highly restricted, leading to reduced self-healing efficiency of the microbial concrete [1]
Pei et al. [2], [3] investigated another spore-forming bacteria, B. subtilis, and experimentally proved that the bacterial cell walls accelerate the carbonation of Ca (OH)2, leading to the precipitation of CaCO3in aqueous medium. Further, Khaliq et al. [4] introduced B. subtilis in concrete using various carrier compounds, namely light weight aggregate and graphite nano platelets. Mondal and Ghosh [5] also prepared microbial concrete using B. subtilis and reported significant improvement in compressive strength, and substantial reduction in water absorption and water permeability of the microbial mortar samples.
- MONDAL, Sandip; GHOSH, Aparna Dey. Spore-forming Bacillus subtilis vis-à-vis non-spore-forming Deinococcus radiodurans, a novel bacterium for self-healing of concrete structures: a comparative study. Construction and Building Materials, 2021, 266, Pp. 121122.
- PEI, Ruoting, et al. Use of bacterial cell walls to improve the mechanical performance of concrete. Cement and Concrete Composites, 2013, 39, Pp. 122-130.
- PEI, Ruoting; LIU, Jun; WANG, Shuangshuang. Use of bacterial cell walls as a viscosity-modifying admixture of concrete. Cement and Concrete Composites, 2015, 55, Pp. 186-195.
- KHALIQ, Wasim; EHSAN, Muhammad Basit. Crack healing in concrete using various bio influenced self-healing techniques. Construction and Building Materials, 2016, 102, . Pp. 349-357
- MONDAL, Sandip; GHOSH, Aparna Dey. Investigation into the optimal bacterial concentration for compressive strength enhancement of microbial concrete. Construction and Building Materials, 2018, 183, Pp. 202-214.
Why did the authors fixed the bacteria’s concentration at 3% with the same case of loading (two-point loads)?
- Authors’ reply: A parametric study is carried out on eight beams which are divided into three groups:
Group (1) discusses the first parameter (bacteria’s concentration), so we have to keep all other parameters constant except the bacteria’s concentration to be able to compare. The bacteria’s concentration is the weight of bacteria to cement weight, and it will differ between 1%, 2% and 3% for the same type of bacteria (Bacillus Subtilis) which is the most popular of all types, and the same case of loading (two-point loads). Then they will be compared to the conventional beam (the control beam without bacteria).
Group (2) discusses the second parameter (type of bacteria). In this group we begin to compare the effect of adding Bacillus Subtilis with the effect of adding other types (E. coli and Pseudomonas sps) with the constancy of bacteria’s concentration and case of loading. The bacteria’s concentration in this group will be constant at 3% with the same case of loading (two-point loads). Then they will also be compared to the control specimen.
Group (3) discusses the third parameter (case of loading) and its effect on the capacity of beams with the constancy of bacteria’s concentration and type of bacteria as 3% of Bacillus Subtilis. Three case of loading will be discussed: one-point load, two-point loads and distributed load on four points.
The authors stated that the numerical curve shows the nonlinearity of the concrete material as it is defined on ANSYS. Why the numerical curve is nonlinear?
- Authors’ reply: Because the concrete is a nonlinear material and is defined on ANSYS as a nonlinear material to be simulated accurately. The concrete has a nonlinear stress-strain curve which expresses their nonlinearity.
Reviewer 2 Report (Previous Reviewer 2)
The authors have properly answered all my questions. The purpose and meaning this work are more clear to the readers in this revised manuscript. I think this paper has met the standard of Materials, so that I recommend it to be published.
Author Response
Thank you
Reviewer 3 Report (Previous Reviewer 3)
The manuscript entitled "Finite Element Analysis of Self-Healing Concrete Beams using Bacteria" presents an experimental study conducted on the modeling of flexural strength behavior of concrete beams with bacteria addition. The article is a resubmission to materials and the authors addressed some of my comments from the initial evaluation. However, the paper needs major revisions before it is processed further, some comments follow:
Abstract: The abstract is written qualitatively. My comments were somehow addressed. However, a quantitative evaluation is still missing. Please clearly state in percentage what was the influence of bacteria on the beam’s capacity. For example: "3% of bacteria result in a 5% increase in beam capacity".
Introduction
The introduction should be significantly improved. Please clearly highlight the pros and cons of previous results and justify the need for the current research. Please discuss the highlights individually and assure a clear correspondence between the affirmations from the manuscript and those from the cited papers (the citations introduced in a bulk form ("[1-4]" “[5-10]”). By removing the bulk form, I wasn’t referring to providing the numbers in ascending order, it is about discussing each study separately. Please provide one short sentence for each manuscript that was cited that show its relevance to the current study.
3. Results and discussions
The standard deviation error should be provided for each measurement. These are multiphase/multicomponent materials, only one value/sample cannot be used for comparative evaluation. Some of the differences can be within the error measurement.
Please approximate all the values from the graph considering the error of the equipment and provide the deviation bar in each graph.
Future directions and limitations: Please provide some future directions and limitations of the study. Consider the low number of samples as one of the limitations of the study and conduct an assessment of the credibility of the numerical model.
Author Response
A new manuscript is attached after modifications, so kindly find the attached file.
The following is a point-by-point response:
- Reviewer comment: Abstract: The abstract is written qualitatively. My comments were somehow addressed. However, a quantitative evaluation is still missing. Please clearly state in percentage what was the influence of bacteria on the beam’s capacity. For example: "3% of bacteria result in a 5% increase in beam capacity".
- Authors’ reply: The required modification has been done.
- Reviewer comment: The introduction should be significantly improved. Please clearly highlight the pros and cons of previous results and justify the need for the current research.
- Authors’ reply: The introduction has been improved. A paragraph from line 105 to 114 explains the pros and cons of previous results and justify the need for the current research. The previous researches studied the mechanical properties only for cubes, cylinders and prismatic specimens of self-healing concrete. While our research is an application of adding bacteria to reinforced concrete elements in a large scale such as beams to study their behavior under the influence of using bacteria.
- Reviewer comment: Please discuss the highlights individually and assure a clear correspondence between the affirmations from the manuscript and those from the cited papers (the citations introduced in a bulk form ("[1-4]" “[5-10]”). By removing the bulk form, I wasn’t referring to providing the numbers in ascending order, it is about discussing each study separately. Please provide one short sentence for each manuscript that was cited that show its relevance to the current study.
- Authors’ reply: The required modification has been done.
- Reviewer comment: The standard deviation error should be provided for each measurement. These are multiphase/multicomponent materials, only one value/sample cannot be used for comparative evaluation. Some of the differences can be within the error measurement. Please approximate all the values from the graph considering the error of the equipment and provide the deviation bar in each graph.
- Authors’ reply: There is no equipment to consider the error and provide the deviation because it is an analytical study on ANSYS software. But we have a tolerance for our simulation which is entered on ANSYS for all models to obtain an accurate solution. This tolerance is equal to 0.05.
- Reviewer comment: Please provide some future directions and limitations of the study. Consider the low number of samples as one of the limitations of the study and conduct an assessment of the credibility of the numerical model.
- Authors’ reply: The future directions and limitations has been added. The credibility of the numerical model has been already discussed in section of the verification models and their results from line 199 to 225.
Round 2
Reviewer 3 Report (Previous Reviewer 3)
The authors addressed all of my comments and the article was improved accordingly. The paper can be processed further.
Author Response
Thank you sir
This manuscript is a resubmission of an earlier submission. The following is a list of the peer review reports and author responses from that submission.
Round 1
Reviewer 1 Report
The abstract does provide any info about the results achieved.
In line 45, the Chemical symbol needs to be corrected.
Figure 1, is not readable.
The whole work is based on the verification of two experimental test results and as such the present work is rather not of any worthwhile reliability.
Reviewer 2 Report
This paper studies the mechanical properties of self-healing concrete beams through experiments and simulations. Some meaningful conclusions are drawn in this paper. However, these experiments and conclusions are not new, which have been widely studied and published. The authors just repeat the similar experiments. Despite of this, some analysis and conclusion are not rigorously described by the author. Following are my remarks for the author to modify their work. For this manuscript, I recommend a rejection.
1. In Line 136, the author claims that there is good agreement between experiment and simulation. However, I can’ agree with the author. Obviously, the simulation result shows that the material performs stiffer in the initial loading stage, and softer when the displacement exceeds 2mm. Then author need describe the phenomenon more accurately.
2. In Line 74, the author claims that the mechanical properties for concrete are assumed to be changed, but the author doesn’t describe how the mechanical properties evolve, which is important to simulation. An explicit expression should be given.
3. In Fig 5, the simulation results are unreasonable, because the experimental loading response demonstrates that the stiffness of materials almost maintains a certain value, however the stiffness of materials simulated by ANSYS gradually decays. The author should carefully check their simulation results, and give a reasonable explanation.
4. Some of the conclusion in Section 4 should be rewritten, because some conclusions can’t be derived by the previous analysis. For example, the first conclusion is doubtable as seen in my first remark.
Reviewer 3 Report
The manuscript entitled "Finite Element Analysis of Self-Healing Concrete Beams using Bacteria" presents an experimental study conducted on modeling of flexural strength behavior of concrete beams with bacteria addition. However, the introduction section includes general affirmation without a quantitative evaluation of previous literature, and many other issues must be addressed. The paper needs major revisions before it is processed further, some comments follow:
Abstract: The abstract is written qualitatively. The majority of the qualitative statements should be modified for quantified result comparisons. Please introduce some short affirmation related to the obtained results (what was the optimum concentration of bacteria, which type of bacteria showed the best results, and what was the difference between the loading case).
Introduction Section
The introduction should be significantly improved. Please conduct a comprehensive and exhaustive study of the previous literature (the latest research, currently the references are old- no references from 2020 and 2022, only one from 2021- not enough, please consider the results presented in these studies: https://doi.org/10.3390/ma14227018, https://doi.org/10.1016/j.jobe.2022.104038). Please clearly highlight the pros and cons of previous results and justify the need for the current research. Please discuss the highlights individually and assure a clear correspondence between the affirmations from the manuscript and those from the cited papers (the citations introduced in a bulk form (“[5-10]”, [13-16], [7-25]) should be removed and a clear relation between the cited studies and the presented information should be provided).
2. Materials and Methods
Test Set-Up for the Experimental Beams – Please increase the clarity of the image, also, please introduce figure labels to indicate the areas of interest for the readers.
Figure 4 is not cited in the manuscript text body, also the content is redundant since the information are already presented in Table 2.
The choosing of bacteria type and concentration seems to have no rationale. Why E.coli and Pseudomonas haven't been tested at concentrations of 1 and 2? Please introduce corresponding comments into the manuscript.
Please specify which type of “%” are considered for bacteria (weight or volume)?
3. Results and discussions
Discussion section. The discussion section is missing. In the discussions section, clear correspondence and comparison between the results of this study and those from the literature should be provided. Please improve.